# Preparation and Properties of Magnesium Cement-Based Photocatalytic Materials

**Yongle Fang [1,2], Chang Shu [1,2], Lu Yang [1,2,*], Cheng Xue [3,*], Ping Luo [3] and Xingang Xu [1,2]**

1   State Key Laboratory of Silicate Materials for Architectures, Wuhan University of Technology, 122# Luoshi Road, Wuhan 430070, China; 290585@whut.edu.cn (Y.F.); shuchang@ramentalk.cn (C.S.); xuxg123@foxmail.com (X.X.)
2   School of Materials Science and Engineering, Wuhan University of Technology, Wuhan 430070, China
3   China Construction Sixth Engineering Bureau Corp., Ltd., Tianjin 300170, China; luo10012022@163.com
*   Correspondence: yanglu@whut.edu.cn (L.Y.); yangziyety@163.com (C.X.)

**Abstract:** Photocatalytic oxidation is a technology developed in recent years for the degradation of indoor air pollutants. In this study, magnesium cement-based photocatalytic material (MPM) was prepared by loading $TiO_2$ photocatalysts onto a $SiO_2$-modified basic magnesium chloride whisker (BMCW) surface, and was subsequently sprayed evenly on the surface of putty powder to form a photocatalytic functional wall coating (PFWC) material. Then, by introducing Ag, visible light photocatalytic functional wall coating (VPFWC) materials were also prepared. The results show that $TiO_2$ and $SiO_2$ form Ti–O–Si bonds on the BMCW surface, and the PFWC presents a promising degradation effect, with a photocatalytic removal rate of 46% for gaseous toluene, under ultraviolet light for 3 h, and an MPM coating concentration of 439 $g/m^2$. This is related to the surface structure of the functional coating, which is formed using putty powder and MPM. The visible light photocatalytic efficiency of the VPFWC increased as the spraying amount of the $AgNO_3$ solution increased, up to 16.62 $g/m^2$, and then decreased with further increasing. The gaseous toluene was degraded by 28% and 73% in 3 h, by the VPFWC, under visible light and ultraviolet light irradiation, respectively. In addition, the photocatalytic performance of the PFWC/VPFWC also showed excellent durability after being reused five times.

**Keywords:** magnesia cement; $TiO_2$; photocatalytic; wall coating; silver modification

## 1. Introduction

It is reported that the average time spent indoors during working is about 80~90% [1,2]. There are many volatile organic compounds (VOCs) in indoor furniture and decorations, which seriously affect people's health and work efficiency. As a consequence, improving indoor air quality and providing a healthy indoor environment have become the focus of attention.

At present, the main removal methods of indoor air pollutants are adsorption [3–6], biodegradation [7,8], photodegradation [9,10], and so on. Although adsorption agents can absorb pollutants, they can only transfer pollutants and, ultimately, cannot eliminate them. There are also some difficulties to be overcome in microbial degradation, such as the complex culture process and long cycle. A microbe needs a suitable environment to survive, which is difficult to apply to indoor environments. Photocatalytic oxidation (PCO), as a promising technology for the degradation of air pollutants, has received extensive attention in past decades [10–14]. As a kind of material with high chemical stability, a low price and high catalytic activity, $TiO_2$ photocatalysts have been widely used in building materials. Combining $TiO_2$ photocatalysts with building materials not only endows the building materials with functionalization, but also provides a new method to solve indoor air pollution [15–17]. A pure $TiO_2$ photocatalyst is difficult to disperse, which easily causes catalyst waste, and the complex environment (containing free ions and a high pH value)

of ordinary Portland cement systems and the encapsulation of hydration products have adverse effects on the catalytic performance of $TiO_2$ [18–20]. In addition, $TiO_2$ can only be excited by ultraviolet light, whose wavelength is less than 387 nm, while sunlight, whose wavelength is less than 387 nm, only accounts for about 4%, which also limits its application [21,22].

As the main component of magnesium oxychloride cement, the basic magnesium chloride whisker (BMCW) has a simple and stable structure, avoiding the complex environment that affects the photocatalysis of $TiO_2$. In addition, the BMCW is crisscrossed to form a special pore structure, which is an excellent catalytic carrier. According to the characteristics of the building materials of indoor decorations, $TiO_2$ sol was loaded onto the $SiO_2$/BMCW surface to prepare the magnesium cement-based photocatalytic material (MPM), by using the BMCW as the loading matrix and $SiO_2$ as the matrix modifier [23]. Then, the MPM was compounded with cement-based putty powder to prepare the photocatalytic functional wall coating (PFWC), which can degrade indoor pollutants. The effects of spraying different amounts of the coating onto the surface structure, and the photocatalytic performance of the materials, were studied. In addition, in order to further expand the application range of the material, a functional wall coating with visible light catalytic activity (VPFWC) was prepared, using Ag as the deposition modification material [24–26]. The effects of the deposition amount of Ag on the photocatalytic performance of the material were studied.

## 2. Results and Discussion

### 2.1. Material Characterization

Figure 1 shows the FT-IR diagram of the MPM, from which we can observe that the absorption peaks at 3681 $cm^{-1}$, 3640 $cm^{-1}$, 3418 $cm^{-1}$ and 1635 $cm^{-1}$ are the stretching vibration peaks of –OH in adsorbed water and basic magnesium chloride, respectively, and 1447 $cm^{-1}$ and 1386 $cm^{-1}$ are the characteristic peaks of –OH in basic magnesium chloride whiskers. The vibrational absorption peaks of the Si–O–Si bonds are 1043 $cm^{-1}$ and 857 cm, and 572 $cm^{-1}$ and 433 $cm^{-1}$ at low wavenumbers are the overlapping absorption peaks of Si–O–Si, Mg–O, Ti–O–Ti and other bonds. Some studies have shown that [12,27] the peak in the range of 900–1000 $cm^{-1}$ is the vibration absorption peak of the Ti–O–Si bond. Moreover, the peak at 974 $cm^{-1}$ indicates that the $TiO_2$ photocatalyst has been successfully supported on the surface of the $SiO_2$/BMCW.

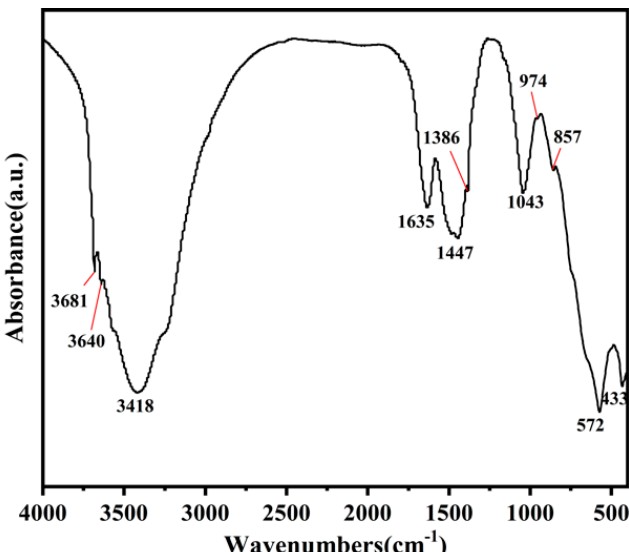

**Figure 1.** FT-IR spectra of magnesium cement-based photocatalytic materials.

Figure 2 shows the microstructure of the BMCW, $SiO_2$/BMCW and MPM. It can be observed from Figure 2a that the BMCWs are crisscrossed with each other, with a good

fibrous morphology, uniform thickness, and no obvious bending phenomenon, indicating that the whisker has good rigidity and uniform distribution. It can be observed from Figure 2b that nano-$SiO_2$ presents the shape of nano-sphericity, without obvious agglomeration, and is evenly distributed along the length and diameter of the whisker, which is conducive to improving the water resistance of the BMCW and increasing the load performance of the $TiO_2$ photocatalytic material [23]. It can also be observed from Figure 2c that nano-$TiO_2$ wraps around the surface of the nano-$SiO_2$/BMCW and forms a close bond, which is consistent with the observation of the Si–O–Ti bond formed in FT-IR spectroscopy (Figure 1). In addition, it can be observed that nano-$TiO_2$ forms a certain agglomeration, which increases the specific surface area of the $SiO_2$/BMCW, enhances the adsorption of pollutants and improves the photocatalytic efficiency of $TiO_2$.

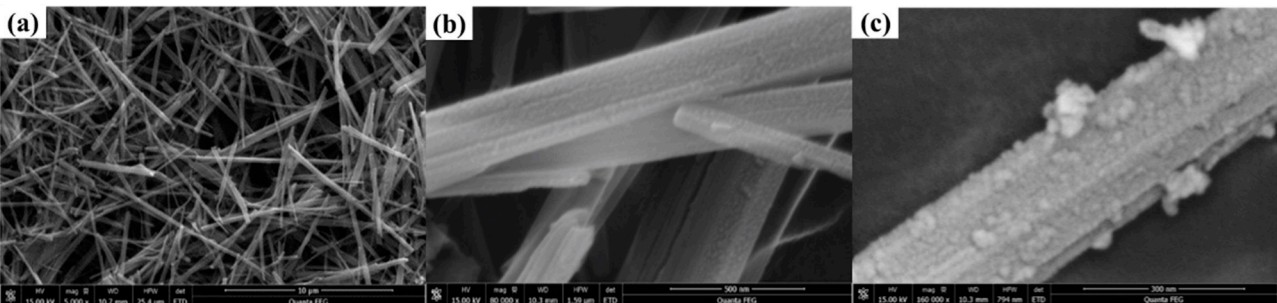

**Figure 2.** SEM images of (**a**) the BMCW, (**b**) $SiO_2$/BMCW and (**c**) MPM.

Figure 3 shows the microscopic morphology of the PFWC. The morphology of the putty powder without the MPM (Figure 3a) is irregular, showing large and flocculent hydration products of different sizes. When the MPM was sprayed on the surface of the putty powder, after hydration for 2 h, a small number of MPM nanorods were observed to "extend" from the putty powder hydration product, while the other part was below the putty powder hydration product, and such MPM nanorods are less likely to appear in the figure (Figure 3b). With the increase in the MPM spraying amount, a large number of MPM nanorods were observed on the surface of the putty powder coating, and were interlaced with the large hydration products of the putty powder (Figure 3c). When the spraying amount of the MPM on the surface of the putty powder reached 439 g/m$^2$, the large hydration products on the surface of the putty powder (Figure 3d) decreased significantly. As the putty powder continued to hydrate, it bonded to the cross-linked MPM nanorods, which formed a surface-connected pore structure, conducive to improving the photocatalytic performance of the PFWC.

Figure 4 shows the EDS spectrum of the PFWC-439/putty powder. The EDS analysis results are shown in Table 1. The EDS spots 1 and 3 are the hydration products of the putty powder. Some Ti atoms still exist at the EDS spots 1 and 3, compared with the hydration products of the putty powder, indicating that the MPM nanorods were covered by the hydration products. This also proves that the hydration process of the putty powder remains for a period of time after spraying the MPM on the surface, and covers part of the MPM.

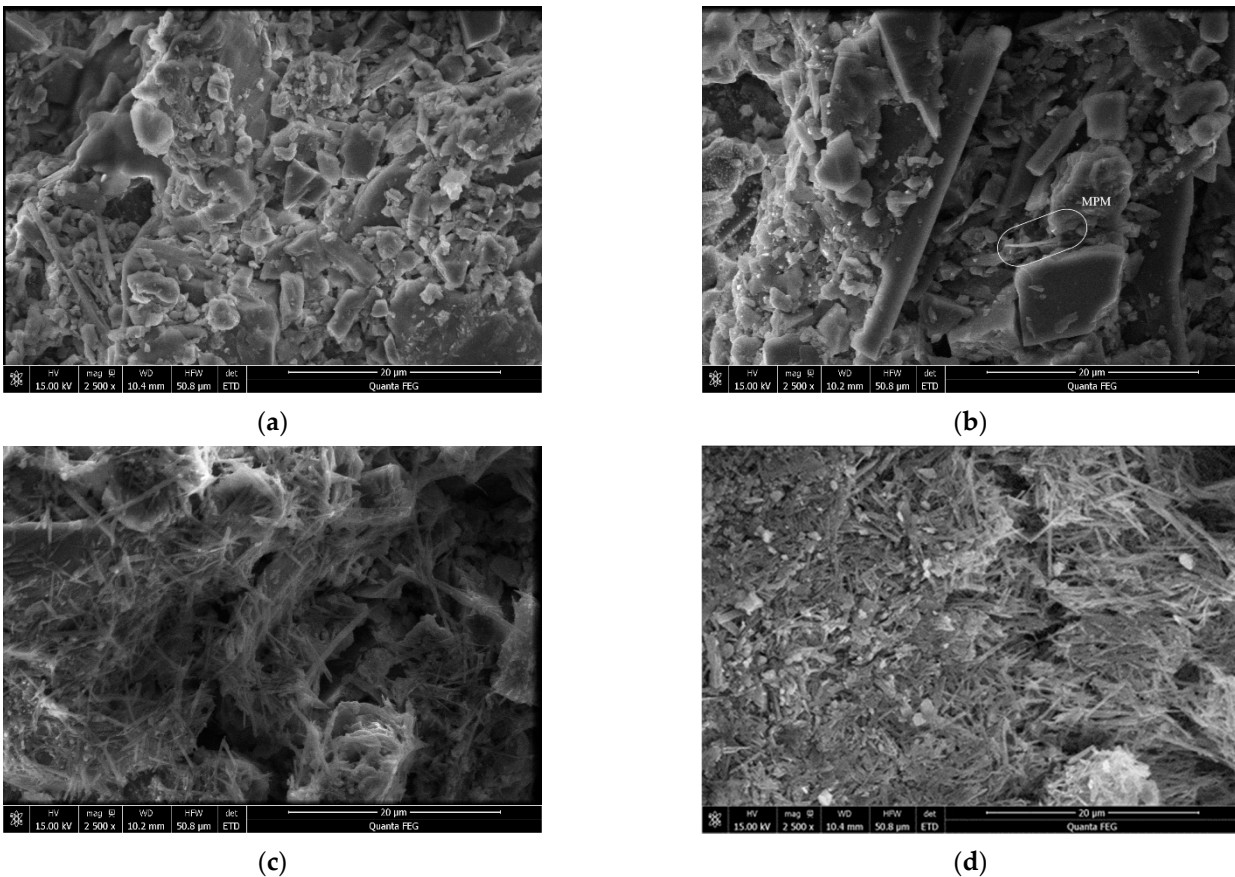

**Figure 3.** SEM images of the morphologies of photocatalytic putty powder with different PFWC spray amounts: (**a**) pure putty powder; (**b**) PFWC-63/putty powder; (**c**) PFWC-133/putty powder; (**d**) PFWC-439/putty powder.

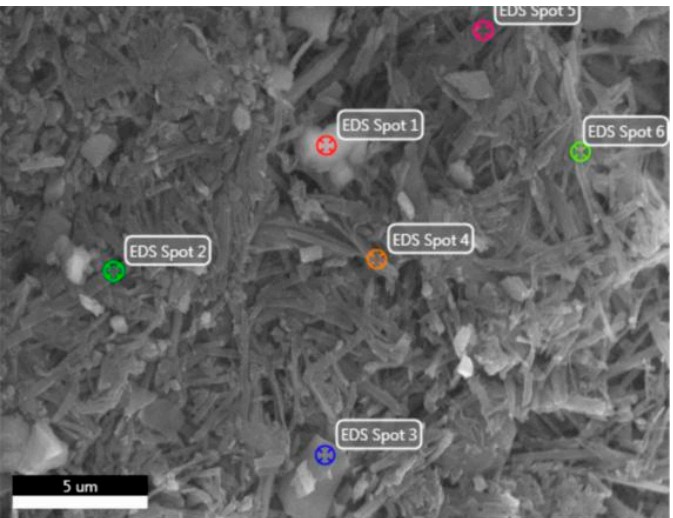

**Figure 4.** EDS image of PFWC-439/putty powder: spot 1 and spot 3 are the hydration products of the putty powder; spot 2, spot 4, spot 5 and spot 6 are MPM.

**Table 1.** EDS analysis of PFWC-439 (at%).

| Element | Spot 1 | Spot 2 | Spot 3 | Spot 4 | Spot 5 | Spot 6 |
|---------|--------|--------|--------|--------|--------|--------|
| O  | 54.76 | 61.76 | 54.03 | 59.56 | 59.27 | 62.53 |
| Mg | 18.27 | 27.96 | 18.05 | 28.81 | 28.31 | 26.97 |
| Al | 0.73  | 0.64  | 0.52  | 0.63  | 0.50  | 0.63  |
| Si | 18.24 | 0.93  | 18.08 | 0.88  | 0.87  | 0.84  |
| Cl | 5.01  | 7.35  | 7.91  | 8.06  | 8.68  | 6.84  |
| Ca | 1.99  | 0.52  | 0.37  | 0.42  | 0.61  | 0.36  |
| Ti | 1.00  | 1.63  | 1.04  | 1.68  | 1.76  | 1.83  |

Figure 5 illustrates the BSEM image of the VPFWC with different amounts of Ag/AgCl surface deposition. The bright point in Figure 5 is the aggregation of Ag/AgCl [24]. These bright spots exist in both the hydration products of the MPM and the putty powder, indicating that Ag/AgCl is uniformly deposited on the surface of the VPFWC-439. The unified deposition of Ag/AgCl plays a positive role in improving the light response range of the VPFWC and the photocatalytic reaction efficiency. Due to the self-aggregation effect of the Ag/AgCl particles, the particle diameter on the surface of the VPFWC became larger and the distribution density increased significantly, with the gradual increase in deposition amount. As shown in Figure 5c, the maximum diameter of Ag/AgCl on the surface of the VPFWC-16.62 reached about 1 µm. When the deposition amount of Ag/AgCl reached 32.35 g/m$^2$, its distribution was more dense on the surface of the VPFWC-32.35 (Figure 5d), but the nanorod-like structure could hardly be observed. It could be that the excessive spraying amount of the AgNO$_3$ solution introduced a large amount of water and anhydrous ethanol, which meant that the MPM was encased in a secondary hydrated putty powder. It can also be observed that the surface of the VPFWC-32.35 is less angular and more rounded than that of the PFWC-439 (Figure 3d).

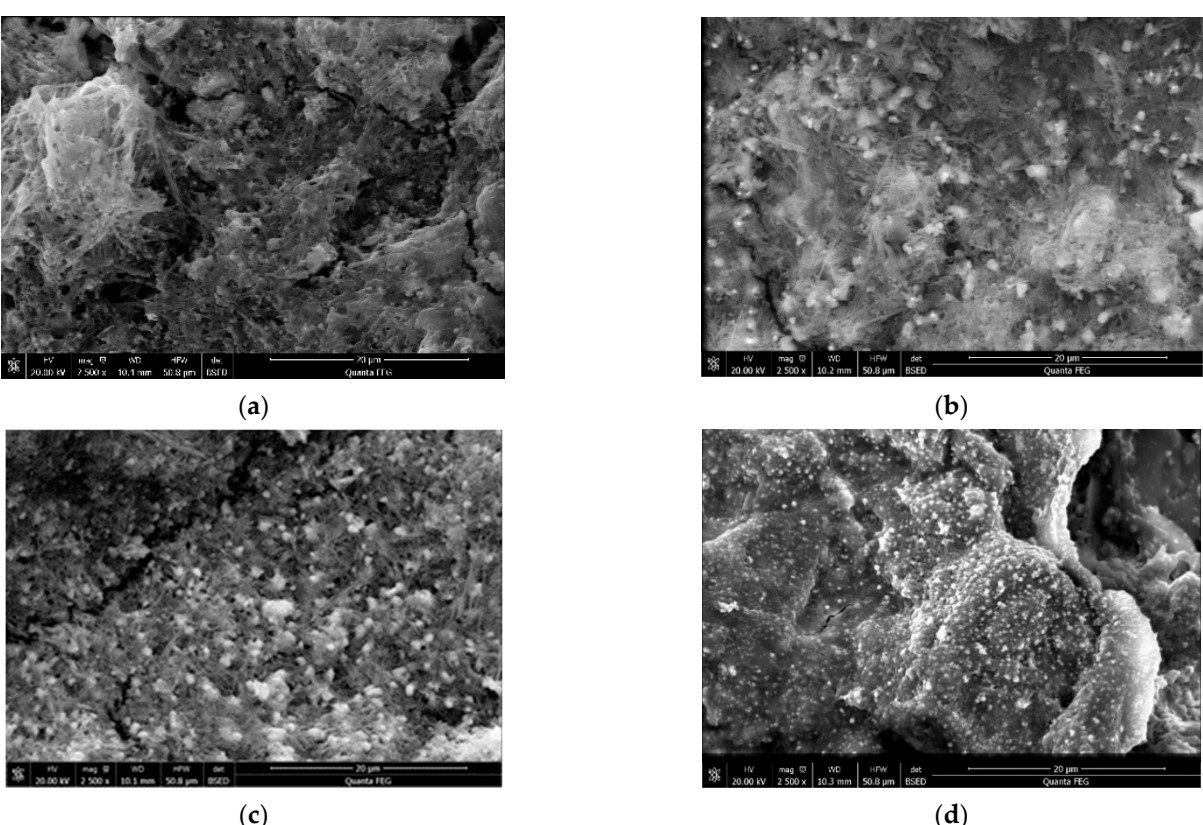

**Figure 5.** BSEM images of different Ag/AgCl surface depositions: (**a**) VPFWC-1.70; (**b**) VPFWC-8.31; (**c**) VPFWC-16.62; (**d**) VPFWC-32.35.

### 2.2. Photocatalytic Performance

Figure 6 shows the ultraviolet photocatalytic degradation efficiency of the PFWC for gas-phase toluene. Compared with the performance of a blank group without sprayed MPM, the photocatalytic efficiency of the PFWC-63 for toluene degradation is low, and the 3 h catalytic efficiency is only 16%. However, with the increase in the MPM spraying amount, the photocatalytic degradation efficiency of the PFWC for toluene gradually increased, and the 3 h catalytic degradation efficiencies of the PFWC-133 and PFWC-439 for gas toluene were 27% and 46%, respectively. This is mainly due to their low photocatalytic efficiency, which is almost wrapped by the hydration products of the putty powder, when the spraying amount is small at the beginning. With the increase in the content of sprayed MPM, the exposure of the MPM on the surface of the putty powder hydration product becomes higher and higher. When the spraying amount reached 439 g/m$^2$, the MPM bonded with the hydrated flocculent and with some small particles to form a pore structure, which increased the specific surface area and permeability, and significantly improved its photocatalytic performance.

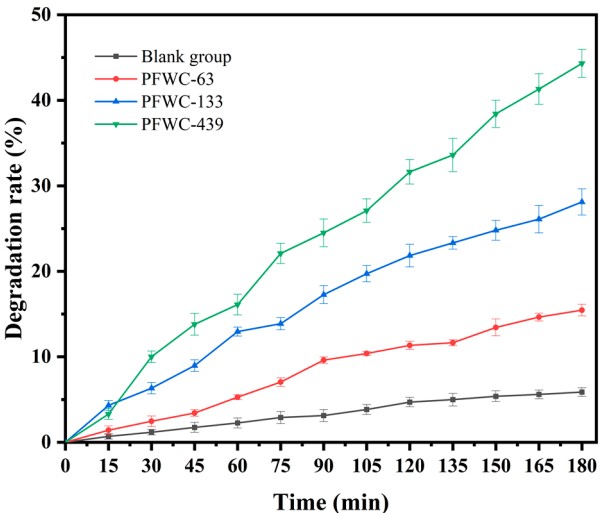

**Figure 6.** The photocatalytic degradation rates of gaseous toluene on the PFWC under ultraviolet light.

In practical applications, the ultraviolet intensity of sunlight is weaker than that of accelerated catalysis in the laboratory, which is a constraint of photocatalytic function on the wall coating. It is particularly important to improve the catalytic efficiency by introducing Ag, which improves the utilization rate of sunlight. Moreover, the Cl$^-$ in the BMCW matrix will self-assemble with Ag$^+$ to form AgCl, which will be partially photolyzed to form a composite structure of Ag/AgCl under visible/ultraviolet light [24].

Figure 7 shows the visible/ultraviolet photocatalytic degradation rate of the PFWC-439 and the VPFWC with different amounts of Ag/AgCl deposition for gas-phase toluene. The results show that the photocatalytic degradation rate of the VPFWC, modified by Ag/AgCl, was higher than that of the PFWC-439 under visible and ultraviolet irradiation. The band gap value of TiO$_2$ is 3.2 eV, which means that only ultraviolet light with a wavelength less than 387 nm can excite TiO$_2$ to generate electron-hole pairs (e$_{cb}$$^-$-h$_{vb}$$^+$, Equation (1)). In addition, the band gap value of TiO$_2$-Ag/AgCl is 2.8 eV [28], which can generate corresponding electron-hole pairs under visible light (e$^-$-AgNPs$^+$, Equation (2)). The electron-hole pairs e$_{cb}$$^-$ and e$^-$ migrate to the surface of the material and react with O$_2$ to generate ·O$_2$$^-$ (Equation (3)), which degrades the target pollutant. At the same time, h$_{vb}$$^+$ and AgNPs$^+$ will migrate to the surface and react with AgCl or the adsorbed OH$^-$ to generate ·Cl$^0$ (Equation (4)) and ·OH (Equation (5)) with strong oxidation ability, and then catalyze the decomposition of the target pollutants. Therefore, VPFWC-modified

Ag deposition shows much higher photocatalytic activity than the PFWC-439 under visible/ultraviolet light catalytic conditions.

$$TiO_2 + UV \rightarrow TiO_2 + e_{cb}^- + h_{vb}^+ \tag{1}$$

$$AgNPs + Vis \rightarrow AgNPs^+ + e^- \tag{2}$$

$$e_{cb}^-/e^- + O_2 \rightarrow \cdot O_2^- \tag{3}$$

$$h_{vb}^+/AgNPs^+AgCl \rightarrow Ag^+ + \cdot Cl^0 + AgNPS \tag{4}$$

$$h_{vb}^+/AgNPs + OH^- \rightarrow \cdot OH + AgNPs \tag{5}$$

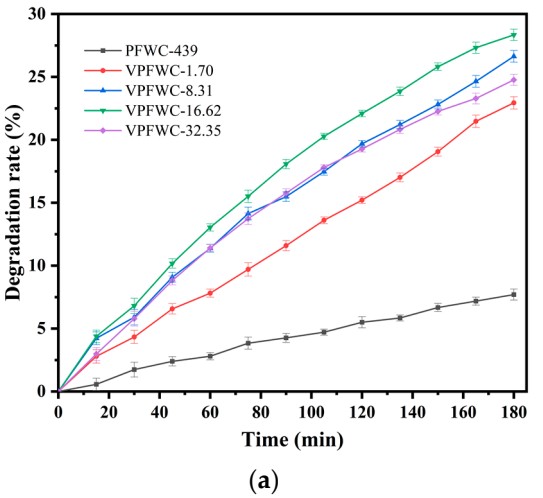
(**a**)

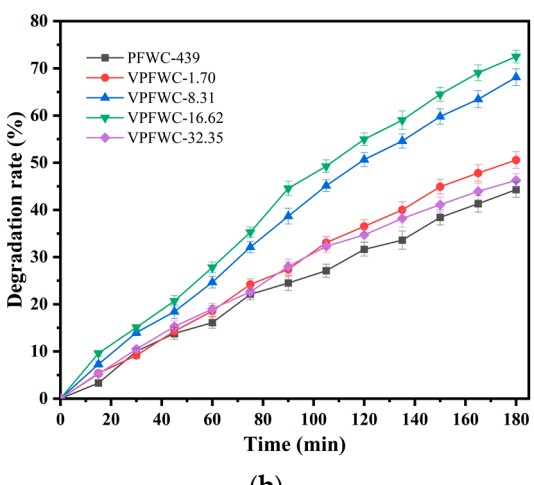
(**b**)

**Figure 7.** The photocatalytic degradation rates of gaseous toluene on the PFWC-439 and the VPFWC-n, with different Ag surface deposits under visible/ultraviolet light: (**a**) visible photocatalytic degradation of the PFWC-439 and the VPFWC-n by visible light; (**b**) photocatalytic degradation of the PFWC-439 and the VPFWC-n by ultraviolet light.

When the deposition amount of Ag/AgCl on the VPFWC surface increased from 1.70 g/m² to 16.62 g/m², its photocatalytic efficiency gradually increased. The photocatalytic degradation rate of gas-phase toluene for the VPFWC-16.62 reached 28% and 73% under visible and ultraviolet light in 3 h, respectively, which was much higher than the 7.5% and 46%, respectively, for the PFWC-439. However, when the deposition amount reached 32.35 g/m², the degradation rate of the VPFWC-32.35 to gas-phase toluene decreased under visible and ultraviolet light catalysis. The degradation rate of the VPFWC-32.35 under ultraviolet light catalysis almost matched that of the PFWC-439. This may be caused by the secondary hydration of the putty powder, induced by water and anhydrous ethanol from the excessively sprayed AgNO₃ solution, which wrapped the PFWC-439 in the secondary hydration product.

### 2.3. Photocatalytic Durability

In order to evaluate the long-term catalytic degradation performance of magnesium cement-based photocatalytic materials in practical applications, the VPFWC-16.62 was selected to carry out a cyclic test of photocatalytic durability. As shown in Figure 8, after five photocatalytic cycle tests, the degradation rate of the VPFWC-16.62 for gas-phase toluene, over 3 h, was stable under both ultraviolet and visible light irradiation, indicating that the photocatalytic durability of the VPFWC-16.62 was excellent.

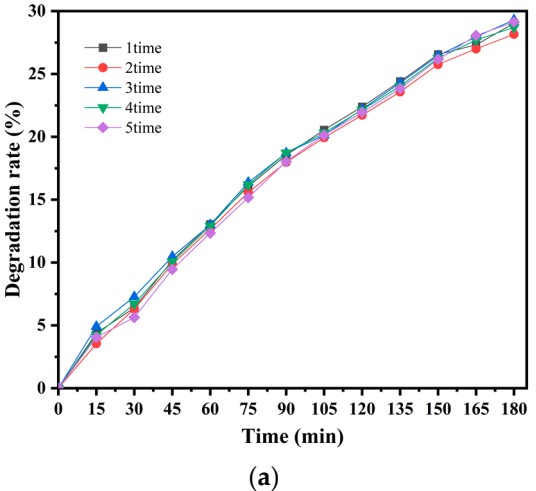
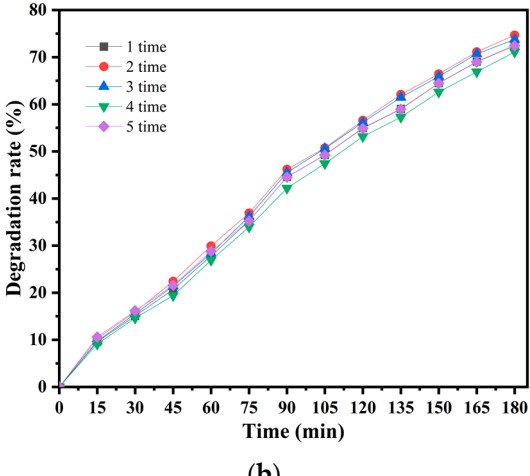

(**a**)  (**b**)

**Figure 8.** Durability test of the VPFWC-16.62 for photocatalytic degradation of gaseous toluene under visible/ultraviolet light: (**a**) photocatalytic removal performance of the VPFWC-16.62 for gas-phase toluene in 5 visible light cycles; (**b**) photocatalytic removal performance of the VPFWC-16.62 for gas-phase toluene in 5 ultraviolet light cycles.

## 3. Materials and Methods

### 3.1. Materials

In this study, magnesium chloride hexahydrate ($MgCl_2 \cdot 6H_2O$, AR), light-burned magnesium oxide (MgO, AR), tetraethyl orthosilicate ($C_8H_{20}O_4Si$, AR), ammonium hydroxide ($NH_3 \cdot H_2O$, AR), titanium sol preparation raw materials of tetra-n-butyl titanate ($C_{16}H_{36}O_4Ti$, AR), nitric acid ($HNO_3$, AR), silver nitrate ($AgNO_3$, AR) and anhydrous ethanol ($C_2H_5OH$, AR) were purchased from Sinopharm Group Chemical Reagent Co., Ltd. Deionized water was prepared using a laboratory EPED-Z1 deionized pure water machine (Shanghai Tims Scientific Instrument Co., Ltd., Shanghai, China). The putty powder, which implemented standard JG/T3049-1998 (industrial grade), was purchased from Shanghai Duoleshi Chemical Coatings Co., Ltd., Shanghai, China.

### 3.2. Synthesis and Characterization

#### 3.2.1. Preparation of Magnesia Cement-Based Photocatalytic Materials

The basic magnesium chloride whisker (BMCW) was prepared according to the molar ratio of n MgO/n $MgCl_2$ = 0.1 [29]; firstly, 4.8 g of light MgO was added to 300 mL of the 4 mol/L $MgCl_2$ solution, then it was stirred at room temperature for 1 h, aged for 48 h, washed and dried at 60 °C.

The preparation of the $SiO_2$-modified basic magnesium chloride whisker ($SiO_2$/BMCW) included the following steps: 1 g of the basic magnesium chloride whisker was added to 80 mL of anhydrous ethanol, then, according to the mass ratio of $SiO_2$/(BMCW + $SiO_2$) = 0.4, 1 mL of TEOS was added to the alcohol solution of the basic magnesium chloride whisker and stirred for 6 h, and 1.5 mL of ammonia was slowly added at the speed of 1~2 gt/s, and then filtered, washed and dried after continuous stirring for 4 h.

The preparation of the magnesium cement-based photocatalysis material (MPM) included the following steps: 14 mL of $TiO_2$ sol [11,12] and 1 g of the above-mentioned $SiO_2$/BMCW were mixed and stirred in 60 mL of anhydrous ethanol for 12 h, then washed with anhydrous ethanol, centrifuged 3 times, and finally dried at 85 °C.

#### 3.2.2. Preparation of Magnesia Cement-Based Photocatalytic Functional Coating

According to the mass ratio of 2:1, 40 g of putty powder and 20 g of $H_2O$ were mixed and stirred evenly, and were placed for 5 min, then the fresh slurry was filled into a mold with a diameter of 10 cm. The demolded samples were cured for 2 h in a YH-40B standard curing box with relative humidity higher than 90% at 25 °C.

After 2 g of MPM and 200 mL of anhydrous ethanol were stirred evenly, according to the concentration of 0.01 g/mL, they were added to the sprayer and uniformly sprayed on the surface of the cured putty powder matrix. According to the actual spraying amount, the spraying amount of the MPM on the surface of the putty powder was 63 g/m$^2$, 133 g/m$^2$ and 439 g/m$^2$, respectively, and was denoted as PFWC-n, n = 63, 133, 439, respectively.

The AgNO$_3$ solution was uniformly sprayed on the optimized photocatalytic coating surface. The proportion of AgNO$_3$:anhydrous ethanol:deionized water was 1:3:1. After curing for 30 min, it was moved to a high-pressure mercury lamp for 10 min to be excited by ultraviolet light. According to the actual spraying amount of AgNO$_3$, the deposition amount of Ag on the coating surface was 1.70 g/m$^2$, 16.62 g/m$^2$, and 32.35 g/m$^2$, respectively, and was denoted as VPFWC-n, n = 1.70, 8.31, 16.62, 32.35, respectively.

### 3.3. Characterization

Fourier transform infrared (FT-IR) spectra were obtained using a Nicolet 6700 Fourier transform infrared spectrometer (Thermo Electron Scientific, Waltham, MA, USA) with 48 scans per sample collected from 400 to 4000 cm$^{-1}$ at 2 cm$^{-1}$ resolution. Surface morphology was observed using an FEI Quanta 450 FEG environmental field emission scanning electron microscope (FEI Quanta 450 FEG, FEI Company, Hillsboro, OR, USA) equipped with Genesis EDS spectroscopy for material compositional analysis. Before testing, the samples were dried at the temperature of 40 °C for 24 h and plated with platinum.

### 3.4. Photocatalytic Performance Test

The characterization of the catalytic performance of the photocatalytic materials was tested using the degradation effect of the simulated pollutant toluene and gas chromatography (GC2020, Wuhan Hengxin Century Technology Co., Ltd., Wuhan, China). The samples with a diameter of 10 cm were put into a 2.35 L stainless steel reactor and 2 μL of toluene was injected into the reactor using a micro syringe. Then, the toluene was volatilized into a gaseous state at 35 °C. We used the gas chromatograph to detect the gas concentration, to ensure that the initial concentration of gaseous toluene in the reactor was about (200 ± 10) ppm. The 300 W high-pressure mercury lamp, with a maximum emission wavelength of 365 nm, was used for ultraviolet light testing, and the 249 W Xenon lamp (CEL-HXF249, Ceaulight Co., Ltd., Beijing, China) was used for visible light testing, by using a 420 nm filter to cut the ultraviolet light. The vertical distance between the lamp and the reactor was adjusted to ensure that the light intensity on the sample surface was 1 mW/cm$^2$. The photocatalytic degradation of toluene can be calculated with Formula (6), as follows:

$$\eta = \frac{C_0 - C}{C_0} \times 100\% \tag{6}$$

In the formula, C$_0$ and C are the initial concentrations of gas-phase toluene and the concentration after the photocatalytic reaction, respectively.

## 4. Conclusions

To solve the problems associated with nano-TiO$_2$ photocatalysts, such as their difficulty dispersing, unsuitability for recycling, and low efficiency of sunlight utilization, the hydration product of the magnesium oxychloride cement (BMCW) was used as the loading matrix to prepare the MPM, which was modified by Ag deposition to expand the material spectral response range and to improve the photocatalytic efficiency. The photocatalytic removal performance of gas-phase toluene and the MPM loaded in putty powder was investigated. The results show that the load performance of TiO$_2$ can be significantly increased by modifying a layer of SiO$_2$ on the surface of the BMCW, resulting in the formation of a Ti–O–Si bond. When the amount of MPM increased to 439 g/m$^2$, MPM nanorods gradually bonded on the surface of the putty powder hydration product, and formed a surface structure containing small holes, which had a good photocatalytic degradation effect on gas-phase toluene. The deposition of Ag can significantly improve the

catalytic efficiency of magnesium cement-based photocatalytic materials. With the increase in Ag deposition, the catalytic efficiency of the materials first increased and then decreased. The VPFWC-16.62 showed the best catalytic degradation efficiency on gas-phase toluene. After five cycles of catalytic tests, it still maintained excellent stability.

**Author Contributions:** Conceptualization, Y.F., C.S. and L.Y.; methodology, Y.F., C.S. and L.Y.; software, Y.F.; validation, P.L.; investigation, Y.F. and C.S.; writing—original draft preparation, Y.F., X.X. and L.Y.; supervision, L.Y. and C.X. All authors have read and agreed to the published version of the manuscript.

**Funding:** This research was funded by the National Natural Science Foundation of China (51802238) and the State Key Laboratory of Solid Waste Resource Utilization and Energy Saving Building Materials Open Fund (SWR-2021-008), and the Fundamental Research Funds for the Central Universities.

**Data Availability Statement:** The data presented in this study are available on request from corresponding author (L.Y.).

**Conflicts of Interest:** The authors declare no conflict of interest.

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
