# Peer review of "Preparation and Properties of Magnesium Cement-Based Photocatalytic Materials"

_catalysts, doi:10.3390/catal12040420_

Round 1
Reviewer 1 Report
Paper deals with the cement material with photocatalytic properties. Concept of the paper and applied methods are adequate and presentation of the results is fairly clear. Still, some parts of methodology are not so clear, so clarification is needed. English also needs thorough check and corrections so I will not mention those errors.
1) In abstract you have abbreviation MPM which is not explained till introduction section. All abbreviations need to be explained when first time used.
2) In introduction part you need to point what is the main novelty of this paper since there is a decent amount of papers dealing with self-cleaning photocatalytic cements.
3) Line 202: “4.8 g light MgO was added…” What is light MgO?
4) Line 207: “mass ratio of SiO2/(…) =40%,…” Ratio can’t be expressed in %.
5) Line 209: “at the speed of 1-2 d/s,…” d is probably drop? Drop is abbreviated gt or gtt (gutta from Latin)
6) What was the setup for the sampling? You just mention that for toluene concertation GC was used. Which one? You need to specify as with the other instruments, Nicolet, FEI etc. Also what type of reactor was used? How exactly was toluene sample taken from reactor? What is the meaning that catalytic performance was “mainly” tested by toluene degradation? Was there some other ways?
7) You use two lamps, Hg and Xe, but for which tests? You have results for visible and UV light. But both of these lamps emit UV and visible light. Did you use some kind of filters to cut UV from visible? You need to explain that.
8) Line 86: You mention that agglomeration of TiO2 causes the increase of the specific surface area. That is not correct, the larger the particle, the smaller is the surface. Agglomeration of TiO2 would cause decrease of surface area.
9) Line 107: “there is still high content of Ti at EDS spot 1 and 3” Those spots have smallest content of Ti according to Table 1.
10) You mention performance of MPM in blank group (line 135), but it is not clear what is the blank group.
11) You should include band gap values of TiO2 and Ag in discussion on photocatalytic effect (Eq 1-5). There is no explanation why is TiO2 excited by UV and Ag by visible light.
Author Response
Dear Editor,
Thank you and the reviewers for their time and thoughtful comments, many of which have been incorporated into the revised manuscript. We added an expanded description of the paper, suggested by reviewers, which makes the manuscript more interesting and informative. All the revised positions were marked in red in paper.
Reviewers’ comments:
1#
Paper deals with the cement material with photocatalytic properties. Concept of the paper and applied methods are adequate and presentation of the results is fairly clear. Still, some parts of methodology are not so clear, so clarification is needed. English also needs thorough check and corrections so I will not mention those errors.
- In abstract you have abbreviation MPM which is not explained till introduction section. All abbreviations need to be explained when first time used.
Thank you for your advice, all abbreviations have been explained when first time used in the revised manuscript.
- In introduction part you need to point what is the main novelty of this paper since there is a decent amount of papers dealing with self-cleaning photocatalytic cements.
Thank you for your comments. As the main component of magnesium oxychloride cement, the basic magnesium chloride whisker (BMCW) has a simple and stable structure, avoiding the complex environment affecting the photocatalysis of TiO2. Also, BMCW are crisscrossed to form a special pore structure, which are an excellent photocatalytic carrier. In this paper, TiO2 sol was loaded on SiO2/BMCW to prepare magnesium phase cement based photocatalytic material (MPM) by using BMCW as loading matrix and SiO2 as matrix modifier. Furthermore, in order to further expand the application range of the material, the functional wall coating with visible light catalytic activity (VPFWC) was prepared by using Ag element as the deposition modification material. we have modified the descriptions in the revised manuscript.
- Line 202: “4.8 g light MgO was added…” What is light MgO?
Thank you for your comments. we have used " light burned MgO " instead of " light MgO " in the revised manuscript. There are two kinds of MgO (light burned MgO and heavy burned MgO). Light burned MgO has high activity and low density compared with heavy burned MgO, which is usually been used to prepare the BMCW, we have modified the description in the revised manuscript.
- Line 207: “mass ratio of SiO2/(…) =40%,…” Ratio can’t be expressed in %.
Thank you for your advice, we have used " mass ratio of SiO2/( BMCW+SiO2) =0.4" instead of " mass ratio of SiO2/(…) =40%" in the revised manuscript.
- Line 209: “at the speed of 1-2 d/s,…” d is probably drop? Drop is abbreviated gt or gtt (gutta from Latin)
We have used " at the speed of 1-2 gt/s " instead of " at the speed of 1-2 d/s" in the revised manuscript.
- What was the setup for the sampling? You just mention that for toluene concertation GC was used. Which one? You need to specify as with the other instruments, Nicolet, FEI etc. Also what type of reactor was used? How exactly was toluene sample taken from reactor? What is the meaning that catalytic performance was “mainly” tested by toluene degradation? Was there some other ways?
Thank you for your advice. The sample with a diameter of 10 cm was put into a 2.35 L stainless steel reactor and 2 μl toluene was injected into the reactor using a micro syringe. Then the toluene was volatilized into a gaseous state at 35°C. We used a gas chromatograph (GC2020,Wuhan Hengxin Century Technology Co., Ltd., Wuhan, China) to detect the gas concentration to ensure that the initial concentration of gaseous toluene in the reactor is about (200±10) ppm. In this paper, the catalytic performance of the sample is characterized by degrading toluene. we have modified the descriptions in the revised manuscript.
- You use two lamps, Hg and Xe, but for which tests? You have results for visible and UV light. But both of these lamps emit UV and visible light. Did you use some kind of filters to cut UV from visible? You need to explain that.
Thank you for your comment. In this paper, the 300 W high pressure mercury lamp with maximum emission wavelength of 365 nm was used for ultraviolet light testing. The xenon lamp is used for visible light testing by using 420 nm filter to cut the ultraviolet light. The vertical distance between the lamp and the reactor was adjusted to ensure the light intensity on the sample surface is 1 mW/cm2. We have added these in the revised manuscript.
- Line 86: You mention that agglomeration of TiO2 causes the increase of the surface area. That is not correct, the larger the particle, the smaller is the surface. Agglomeration of TiO2 would cause decrease of surface area.
Thank you for your comment. We have revised this in the manuscript.
- Line 107: “there is still high content of Ti at EDS spot 1 and 3” Those spots have smallest content of Ti according to Table 1.
Thank you for your comment. We have revised this in the manuscript.
- You mention performance of MPM in blank group (line 135), but it is not clear what is the blank group.
Thank you for your comment. The blank group is putty powder without spraying MPM.
- You should include band gap values of TiO2 and Ag in discussion on photocatalytic effect (Eq 1-5). There is no explanation why is TiO2 excited by UV and Ag by visible light.
Thank you for your advice. The band gap values of TiO2 is about 3.2 eV, which means that only ultraviolet light with a wavelength less than 387 nm can excite TiO2 to generate electron-hole pairs (ecb--hvb+,Equation 1). And the band gap values of TiO2-Ag/AgCl is 2.8 eV, which can be generates corresponding electron-hole pairs under visible light (e--AgNPs+,Equation 2). we have modified the descriptions in the revised manuscript.

Reviewer 2 Report
This paper titled “Preparation and Properties of Magnesium Cement Based Photocatalytic Materials” presents a systematic study on a well-known topic: TiO2-based Photocatalytic Cementitious Composites. In this study, to overcome the classic difficulty concerning the nano-TiO2 photocatalyst dispersing, the hydration product method was used to obtain the quite homogeneous TiO2 NPs decorated magnesium cement based photocatalysis material. The photocatalytic performance was improved by introducing Ag element. The synthesized photocatalytic materials showed a good sustainability. From a general point of view, this paper is interesting, but there are some points to be imperatively improved before to be considered to publish in Catalysts.
- The authors should give the complete definition for all abbreviations, included in Abstract, when they are employed for the first time.
- Please revise the abstract and give the more precis text. For ex. “…the photocatalytic removal rate of toluene reached 46% in 3 h when 18 amount of the MPM composite was 439 g/m2.” One cannot guess this efficiency was obtained under which kind of irradiation.
- In the section of “2.1. Physical and Chemical Propertie”, the text should be revised, because of its didactic style. (BTW, the subtitle “Propertie” should be added a “s”)
- In photocatalysis experience, the auteurs should precis the gas volume to be degraded and the sample size (with a fixed initial gas concentration for all tests), without those information, the photocatalytic efficiency cannot be evaluated and/or compared with other works.
- For all photocatalysis tests, the preliminary experiments should be carried out: i) materials adsorption rate; and ii) photolysis effect (without photocatalysts). The net photodegradation efficiency = total degradation – adsorption – photolysis effect.
- Always for the photocatalysis test: the authors should not only precis the kind of used lamps, but also give the principal wavelength (special for UV lamp); and the more important is to give the irradiation intensity on the sample surface level for each kind of lamps, then, this value should be fixed for all tests.
- All experimental measurements should be presented with error bars.
- In the text and in the equations, the radical symbol is not standard (dark dots at higher position), please rectify them.
- Please put a space before and after “=”, “+/-“, …
Author Response
Dear Editor,
Thank you and the reviewers for their time and thoughtful comments, many of which have been incorporated into the revised manuscript. We added an expanded description of the paper, suggested by reviewers, which makes the manuscript more interesting and informative. All the revised positions were marked in red in paper.
Reviewers’ comments:
2#
This paper titled “Preparation and Properties of Magnesium Cement Based Photocatalytic Materials” presents a systematic study on a well-known topic: TiO2-based Photocatalytic Cementitious Composites. In this study, to overcome the classic difficulty concerning the nano-TiO2 photocatalyst dispersing, the hydration product method was used to obtain the quite homogeneous TiO2 NPs decorated magnesium cement based photocatalysis material. The photocatalytic performance was improved by introducing Ag element. The synthesized photocatalytic materials showed a good sustainability. From a general point of view, this paper is interesting, but there are some points to be imperatively improved before to be considered to publish in Catalysts.
- The authors should give the complete definition for all abbreviations, included in Abstract, when they are employed for the first time.
Thank you for your advice, all abbreviations had to be explained when first time used in the revised manuscript.
- Please revise the abstract and give the more precis text. For ex. “…the photocatalytic removal rate of toluene reached 46% in 3 h when amount of the MPM composite was 439 g/m2.” One cannot guess this efficiency was obtained under which kind of irradiation.
Thank you for your advice. we have used " the photocatalytic removal rate of toluene reached 46% under ultraviolet light for 3 h as the MPM coating concentration was 439 g/m2" instead of " the photocatalytic removal rate of toluene reached 46% in 3 h when amount of the MPM composite was 439 g/m2" in the revised manuscript. We also modified the other descriptions as well.
- In the section of “2.1. Physical and Chemical Propertie”, the text should be revised, because of its didactic style. (BTW, the subtitle “Propertie” should be added a “s”)
Thank you for your advice. we have used "Material Characterization" instead of " Physical and Chemical Properties" in the revised manuscript.
- In photocatalysis experience, the auteurs should precis the gas volume to be degraded and the sample size (with a fixed initial gas concentration for all tests), without those information, the photocatalytic efficiency cannot be evaluated and/or compared with other works.
Thank you for your comment. The sample with a diameter of 10 cm was put into a 2.35 L stainless steel reactor and 2 μl toluene was injected into the reactor using a micro syringe. Then the toluene was volatilized into a gaseous state at 35 °C. We used a gas chromatograph (GC2020,Wuhan Hengxin Century Technology Co., Ltd., Wuhan, China) to detect the gas concentration to ensure that the initial concentration of gaseous toluene in the reactor is about (200±10) ppm. we have modified the descriptions in the revised manuscript.
- For all photocatalysis tests, the preliminary experiments should be carried out: i) materials adsorption rate; and ii) photolysis effect (without photocatalysts). The net photodegradation efficiency = total degradation – adsorption – photolysis effect.
Thank you for your advice. In this paper, the degradation efficiency of the blank group is the rate of photolysis effect. And all samples were tested for the catalytic performance after absorption of the material, we have added this in the revised manuscript.
- Always for the photocatalysis test: the authors should not only precis the kind of used lamps, but also give the principal wavelength (special for UV lamp); and the more important is to give the irradiation intensity on the sample surface level for each kind of lamps, then, this value should be fixed for all tests.
Thank you for your comment. In this paper, that the 300 W high pressure mercury lamp with maximum emission wavelength of 365 nm was used for ultraviolet light testing. The xenon lamp is used to filter the wavelength before 420 nm with a filter, which is used for visible light testing. We adjusted the vertical distance between the lamp and the reactor to acquire light intensity on the sample surface with 1 mW/cm2. we have modified the descriptions in the revised manuscript.
- All experimental measurements should be presented with error bars.
Thank you for your advice. We had added the error bars in the revised manuscript.
- In the text and in the equations, the radical symbol is not standard (dark dots at higher position), please rectify them.
Thank you for your advice, we had rectified the radical symbol in the revised manuscript.
- Please put a space before and after “=”, “+/-“, …
Thank you for your advice, we had put a space before and after “=”, “+/-“in the revised manuscript.

Round 2
Reviewer 1 Report
All week points of the manuscript were adequately addressed by authors, all mistakes corrected so I can support the publication in this form.
Reviewer 2 Report
The authors have done the effort to improve the manuscript’s quality.
This version is much better and could be published in Catalysts.
Some minor modifications could be done during final edition :
- The radical symbol.
- Line 213: space before and after “=”.
This manuscript is a resubmission of an earlier submission. The following is a list of the peer review reports and author responses from that submission.